# Consumption of Non-Nutritive Sweetener during Pregnancy and Weight Gain in Offspring: Evidence from Human Studies

**DOI:** 10.3390/nu14235098

**Published:** 2022-12-01

**Authors:** Guowei Li, Ruoting Wang, Changfa Zhang, Likang Li, Jingyi Zhang, Guiju Sun

**Affiliations:** 1Center for Clinical Epidemiology and Methodology (CCEM), Guangdong Second Provincial General Hospital, Guangzhou 510317, China; 2Father Sean O’Sullivan Research Centre, St. Joseph’s Healthcare Hamilton, Hamilton, ON L8N 4A6, Canada; 3Department of Health Research Methods, Evidence, and Impact (HEI), McMaster University, Hamilton, ON L8S 4L8, Canada; 4Key Laboratory of Environmental Medicine and Engineering of Ministry of Education, Department of Nutrition and Food Hygiene, School of Public Health, Southeast University, Nanjing 210009, China

**Keywords:** non-nutritive sweetener, pregnancy, offspring, obesity, childhood

## Abstract

The relationship between the consumption of maternal non-nutritive sweeteners (NNS) during pregnancy and the risk of obesity in offspring remains inconsistent. We aimed to systematically evaluate and clarify the relationship between NNS intake during pregnancy and weight gain in offspring based on evidence from population and clinical research. Databases including PubMed (via Medline), EMBASE, and the Cochrane Library were systematically searched for eligible human studies. The primary outcome was the differences in body mass index (BMI) z-scores between offspring at 1 year of age who were with and without NNS intake during pregnancy or between offspring with different NNS intake levels during pregnancy. A random-effects meta-analysis was conducted for data synthesis to calculate the weighted mean difference (WMD). A total of six prospective cohort studies were eligible for inclusion, among which three were used for pooled analysis of the BMI z-score. A significant increase was found in an offspring’s weight at 1 year of age in the NNS group when compared with the control group: WMD in BMI z-score = 0.19 (95% CI: 0.07, 0.31), *p*-value = 0.002. Results from the dose-response analysis showed a linear relationship between NNS intake during pregnancy and WMD at 1 year of age: beta = 0.02 (95% CI: 0.001, 0.04) for per serving/week increase in NNS consumption. The whole body of evidence for the review was rated as low quality. In summary, maternal NNS intake during pregnancy was found to be associated with increased weight gain in offspring based on evidence from human studies. Further well-designed and adequately powered studies are needed to confirm this relationship.

## 1. Introduction

The prevalence of obesity has substantially increased in children over the past several decades, with over 20% of children being overweight or obese worldwide [1]. In addition to some well-recognized risk factors including unhealthy diet and decreased physical activities, studies have also demonstrated that prenatal exposure to excess sugar intake may contribute to an increased risk of obesity in offspring [2,3,4,5]. To reduce their overall sugar consumption, the replacement of non-nutritive sweeteners (NNS) with low or no calories for sugar intake could provide pregnant women with an alternative option for sweet-tasting food or beverages that have limited or no calories. Nevertheless, current evidence from population studies has reflected an inconsistent conclusion regarding the relationship between consumption of maternal NNS during pregnancy and the risk of obesity in offspring. For example, while some studies found a significant relationship between NNS intake during pregnancy and the elevated risk of obesity in offspring [6,7] another study did not reveal a significant association between maternal diet soda consumption and childhood adiposity [8]. Additionally, the US Institute of Medicine did not make a specific recommendation for pregnant women but cautioned against the use of NNS in children, emphasizing the lack of evidence for the NNS intake in pregnancy and its effect on offspring [9].

Therefore, in this study, we aimed to systematically evaluate and clarify the relationship between NNS intake during pregnancy and weight gain in offspring based on evidence from population and clinical research. We also critically appraised the methodological issues of the included studies to provide some helpful evaluation for future research directions and potential public health recommendations.

## 2. Methods

We conducted a systematic review and methodological appraisal for the association between the consumption of NNS during pregnancy and obesity risk in offspring from human studies. This study was registered in PROSPERO (Prospective Register of Ongoing Systematic Reviews; identifier: CRD42022361764).

### Search Strategy and Study Eligibility Criteria

Electronic databases including PubMed (via Medline), EMBASE, and the Cochrane Library were systematically searched through 20 September 2022. Synonyms for NNS, pregnancy, and children were used in the search (Appendix A shows the detailed search terms). Reference lists of the included studies and relevant reviews or commentaries were also searched to further include potentially relevant studies. Only human studies published in English were eligible.

We included observational and interventional studies exploring NNS intake during pregnancy in relation to weight gain in offspring. Studies were not eligible for inclusion if they did not provide information on quantified associations between maternal NNS intake during pregnancy and weight gain in offspring. If data from the same participants were reported in multiple studies, we chose the study with the largest sample size for inclusion. We excluded reviews, commentaries, conference papers, and studies published only in abstract form because no sufficient data could be collected. Two reviewers (G.L. and R.W.) independently screened and determined studies for inclusion in duplicate, with any disagreement resolved by discussion. A third reviewer (L.L.) was consulted if no consensus could be reached between the two reviewers.

## 3. NNS and Outcomes

We collected detailed data on natural and artificial NNS intake during pregnancy, where information on NNS may be reported as continuous or categorical variables from the included studies. The primary outcome was the differences in body mass index (BMI) z-scores between offspring with and without NNS intake during pregnancy or between offspring with different NNS intake levels during pregnancy, where BMI was calculated as weight (in kg) divided by height (in m^2^) and the BMI z-scores were age and sex specific. If the outcomes were reported as multiple points corresponding to different ages of offspring, we used the data on infancy at 1 year of age or those nearest to the infancy at 1 year of age for the main analyses. Secondary outcomes included differences in waist circumference (in cm) and body fat percentage (in %) between offspring.

## 4. Data Extraction and Assessment of Study Quality for Individual Studies

Two reviewers (G.L. and R.W.) independently extracted information from the included studies, in which the collected data covered general information (authors, journal and publication year, country, and study design), participants (age and number of the included pregnant women and offspring, sex of offspring, baseline BMI and diabetes mellitus for pregnant women, and smoking and intake of sugar-sweetened beverage for pregnant women), NNS (type, quantity, frequency, duration, and measurement method), outcomes, study duration, and measures of associations. Some results displayed in graphic form were extracted by using the GetData Graph Digitalizer (http://getdata-graph-digitizer.com; version 2.26.0.20 (accessed on 2 October 2022)).

We evaluated the individual study quality by using Cochrane Collaboration ROBINS-I (the Risk Of Bias In Non-randomized Studies of Interventions) for observational studies [10] and the RoB (Risk of Bias) tool for interventional studies [11]. Each observational study was rated as either low, moderate, or serious risk of bias, while each interventional study could be graded as high, unclear, or low risk of bias.

## 5. Statistical Analyses

All analyses were conducted using the STATA (Stata Corp., College Station, TX, USA; version 14.0) and R (R Foundation for Statistical Computing, Vienna, Austria; version 3.5.1) packages. We used a random-effects meta-analysis to pool the data on outcomes. The pooled associations were calculated and presented as either adjusted weighted mean difference (WMD) for continuous data or adjusted odds ratio (OR), relative risk (RR), or hazards ratio (HR) for categorical outcome data, respectively. Each measure of association was followed by a 95% confidence interval (CI). For the offsprings’ outcomes, we performed a pooled analyses comparing their mothers, between those consuming NNS versus those without NNS intake (i.e., ‘yes’ vs. ‘no’) if the included studies reported results of this single pair-wise analysis or between those consuming the highest level of NNS versus those with the lowest NNS intake level (i.e., ‘highest’ vs. ‘lowest’) if the included studies reported data on multiple NNS intake groups.

Heterogeneity among the included studies was evaluated by using both the Q test and the *I*^2^ statistic, where a *p*-value < 0.1 or the *I*^2^ > 50% indicated significant heterogeneity [12]. Subgroup analyses were conducted for heterogeneity exploration, including the type of NNS (nature vs. artificial), quantity of NNS intake (where the median was used to group pregnant women as either with high or low NNS intake), duration of NNS intake (using the median for dichotomization of long or short duration), age (at birth, early childhood, and mid-childhood), and sex (boy vs. girl) of offspring. We also performed two sensitivity analyses to test the robustness of the main results including (1) using a fixed-effects meta-analysis and (2) excluding studies with a high or serious risk of bias from analyses.

If there were no less than three studies reporting different levels of NNS intake in relation to outcomes, or different results for NNS intake related to weight gain in offspring at different ages, we performed a dose-response analysis with a linear model for the relationship between NNS intake during pregnancy and weight gain in offspring. Furthermore, we conducted the methodological appraisal of the included studies and present the relevant findings qualitatively.

## 6. Evaluation of Publication Bias and Quality of a Whole Body of Evidence

We evaluated the potential publication bias by using the Begg’s test and the funnel plot for the primary outcome. The GRADE (Grading of Recommendations, Assessment, Development, and Evaluation) tool was used to assess the quality of a whole body of evidence across the included studies, in which the whole body of evidence could be rated as either high, moderate, low, or very low quality [13].

## 7. Results

A total of 1251 records were identified from the databases (451, 767, and 33 from PubMed, EMBASE, and Cochrane Library, respectively), among which 79 full-text studies were evaluated for eligibility (Figure 1). We included six cohort studies for analyses: two from the CHILD Birth Cohort (Canada) with a sample size of 2413 and 2298 mother–singleton child dyads [6,14], two Project Viva (US) with a sample size of 1683 and 1078 [7,8], one Danish National Birth Cohort (Denmark) with a sample size of 918 [15], and one Perined-Lifelines Birth Cohort (the Netherlands) with a sample size of 1698 [16]. We used data from the studies with the larger sample sizes for the main analyses (i.e., the publication in JAMA Pediatrics for CHILD Birth Cohort [6] and the publication in International Journal of Obesity for Project Viva [7], respectively); however, the other publication from the same cohort was also used for secondary analysis. Subsequently, we present characteristic descriptions of the four included studies in Table 1. The four studies with 6712 mother–singleton child dyads were published between 2016 and 2022 and with the design of a prospective cohort study. The mean age of mothers ranged from 29 to 33 years, and their BMI ranged from 24 to 28 kg/m^2^. All the NNS were artificial and measured by a validated food frequency questionnaire (FFQ). There were three studies reporting data on BMI z-scores in offspring. Two studies were rated as moderate risk of bias due to a lack of information on missing data [7,15], while the others were graded as low risk of bias.

As shown in Figure 2a and Table 2, a pooled analysis from the three studies yielded a significant increase in an offspring’s weight at 1 year of age in the NNS group when compared with the control group: WMD in BMI z-score = 0.19 (95% CI: 0.07, 0.31), *p*-value = 0.002. No significant heterogeneity was found (*p*-value = 0.83, *I*^2^ = 0%). Table 2 also displayed the results for BMI z-score WMD in offspring at different ages: WMD = 0.06 (95% CI: −0.11, 0.24; *p*-value = 0.48) for offspring at birth (Figure 2b); WMD = 0.20 (95% CI: 0.07, 0.33; *p*-value = 0.002) for offspring at early childhood (Figure 2c); and WMD = 0.29 (95% CI: 0.12, 0.46; *p*-value = 0.001) for offspring at mid-childhood (Figure 2d). There was no significant heterogeneity observed for an offspring’s WMD at early childhood.

Due to the insufficient number of included studies, no subgroup analyses by type/quantity/duration of NNS intake could be conducted. Additionally, there were no studies reporting data on the difference in waist circumferences between offspring. Only one study demonstrated a significantly higher sum of skinfolds in offspring with the highest level of maternal NNS intake during pregnancy; however, there was no significant difference in fat mass index for offspring between the highest and lowest levels of maternal NNS intake during pregnancy [7]. One study reported a significant relationship between the intake of artificial NNS products and increased birth weight; nevertheless, these data were not used for a pooled analysis due to a lack of estimation of BMI that accounted for birth height [16]. There were two studies displaying results for NNS intake during pregnancy and the risk of offspring being overweight or obese at 1 year of age: RR = 0.90 (95% CI: 0.39, 1.91) from the Danish National Birth Cohort [15] and OR = 2.19 (95% CI: 1.23, 3.88) from the CHILD Birth Cohort [6]. We did not synthesize these data on a categorical outcome of an offspring’s weight because of insufficient information to transfer the association measure of RR to the adjusted OR (or vice versa).

The results from the sensitivity analysis by using the fixed-effects model displayed a significant relationship between NNS intake during pregnancy and an increased BMI z-score for offspring at 1 year of age (WMD = 0.19, 95% CI: 0.07, 0.31).

Appendix A displays scatter plots for the different levels of NNS intake during pregnancy in relation to WMD in offspring at 1 year of age. The results from the dose-response analysis showed a linear relationship between NNS intake during pregnancy and WMD at 1 year of age: beta = 0.02 (95% CI: 0.001, 0.04), *p* = 0.037 for per serving/week increase in NNS consumption (Figure 3a). However, there was no significant dose-response relationship between offspring’s elevated ages and WMD, where the WMD was between the offspring with and without maternal NNS intake or between offspring with the highest and lowest level of maternal NNS intake: beta = 0.004 (95% CI: −0.003, 0.011; *p* = 0.27) for a per-1-year increase in an offspring’s age (Appendix A; Figure 3b).

No statistically significant publication bias was found (*p* = 0.60 for Begg’s test; Appendix A). The whole body of evidence across the included studies was rated as low due to the observational study design and lack of the factors that could increase the evidence quality.

Table 3 presents some methodological issues identified from the included studies by study design factor. For instance, no predefined subgroup analyses by mothers’ gestational diabetes were performed to explore whether the relationship between maternal NNS intake and an offspring’s weight gain depended on gestational diabetes [6,7]. One study only assessed the consumption of diet soda, thereby underestimating the intake of NNS from other beverages or foods [8]. There were various uses of the comparators including never [14,15], lowest frequency [6,8], and the first quartile [7] of NNS intake during pregnancy, yielding the heterogeneity of defining a control group.

## 8. Discussion

In this study, we systematically assessed the relationship between maternal NNS intake during pregnancy and weight gain in offspring. A significantly increased WMD in BMI z-score was found in offspring at 1 year of age with NNS intake during pregnancy when compared with control. There was a significant dose-response relationship between NNS intake and increased WMD in offspring at 1 year of age. Nevertheless, the whole body of evidence for this current review was graded as low quality.

NNS has been increasingly used by pregnant women. For example, the prevalence of NNS intake in US pregnant women increased from 16% in the years 1999–2004 to 24% in 2007–2014 based on data from the National Health and Nutrition Examination Survey [17]. NNS intake during pregnancy was reported to have adverse effects on an offspring’s metabolism and microbiome in animal studies [18,19]. Potential mechanisms, including the early-in-life programming of metabolism and taste preferences for offspring, the altered gut microbiota, and the dysregulation of hormone secretion, may account for the link between NNS intake during pregnancy and an increased risk of obesity in offspring [20,21]. However, the evidence from human studies remains sparse and limited. Our systematic review found a significant association between maternal NNS intake during pregnancy and elevated WMD in BMI z-score for offspring, which could generate some insights into the cautious use of NNS for pregnant women. Nonetheless, the small number of included studies and the low quality of the whole body of evidence requires more research for further confirmation and exploration.

There was no significant association between NNS intake during pregnancy and BMI z-score of offspring at birth (Table 2), indicating the apparent impact of maternal NNS consumption on weight gain later in life (rather than fetal growth) and, thus, the potential latency between exposure during pregnancy and obesity phenotype [22]. The results from the dose-response analysis for the increased BMI z-score in offspring with elevated ages, albeit non-significant, may also partly support the potentially more potent effect of early-in-life programming on an offspring’s obesity later in life (Figure 3b). Two studies reported a more prominent relationship between NNS intake during pregnancy and offspring being overweight or obese in boys when compared with girls (RR of 2.31 vs. 1.64 [15] and OR of 3.07 vs. 0.45 [6]). The difference in offspring sex may be due to the variance in gut microbiota and predisposition to metabolic disorders between boys and girls [23,24]. However, because of the limited number of included studies and the heterogeneity of the association measures reported, we could not further explore whether there was a significant subgroup effect of sex on the relationship between NNS intake during pregnancy and weight gain in offspring.

Accumulating studies have been performed to investigate the use of NNS in relation to metabolic syndrome, gestational diabetes in pregnancy, cancers, and infertility [25,26,27,28]. One recent systematic review based on animal studies suggested a small decrease in offspring birth weight in the NNS group when compared with a control group [29]. However, no systematic review on human studies of NNS for weight gain in offspring was available. Cai et al. [30] conducted a meta-analysis to assess the relationship between NNS intake during pregnancy and birth weight by including three cohort studies [6,15,31]. Nevertheless, several methodological defects existed in their review. First, the two included studies shared the same participants from the Danish National Birth Cohort (1999–2002) [15,31], which would yield a spurious estimate of the pooled analysis by using duplicate information. Additionally, the use of birth weight as an outcome may not be a convincing choice due to the lack of consideration of birth height. Moreover, the data on birth weight they extracted were indeed a covariate reported from the included studies [6,15,31]; therefore, the raw comparisons of birth weight without any adjustment between the NNS and control groups would be prone to a confounding bias. Interestingly, if the data of interest were birth weight as a covariate, they missed an eligible interventional trial [32] that assessed the effect of NNS on mutans streptococci but presented data on birth weight as a covariate, which was pointed out in a previous narrative review [19]. Thus, these methodological issues would substantially weaken the strength of their results regarding increased birth weight in the NNS group during pregnancy. By contrast, our review used adjusted differences in BMI z-score as the outcome after performing an extensive search in duplicate and observed an elevated WMD for offspring in the NNS group with rigorous analyses. These findings may help emphasize the hypothesis about the effect of maternal NNS intake on a postnatal obesity risk, which, if further externally confirmed and validated, would inform the public regarding the careful use of NNS during pregnancy.

This study was the first to systematically explore maternal NNS consumption during pregnancy in relation to an offspring’s weight gain, to the best of our knowledge. Study processes including search, data extraction, analysis, and quality assessment were conducted in duplicate to ensure accuracy. The methodological appraisal emphasized the lack of solid evidence of the NNS intake in relation to an offspring’s weight gain and provided some future research directions to improve the quality, reproducibility, and credibility of further human studies, which may help translate the future synthesized evidence into public health policy making and recommendations. Some limitations existed in our review. First, the insufficient number of included studies precluded us from further explorations and syntheses. For instance, we tried to collect the adjusted relationship between NNS and an offspring’s weight gain from the published human studies. However, the covariate adjustment in the models may not be the same as from all the included studies, while some studies may fail to account for important maternal risk factors including socio-economic status, breastfeeding, gestational diabetes, mode of delivery, and maternal BMI. Unfortunately, we could not perform these subgroup analyses stratified by the maternal risk factors due to the insufficient included studies and limited data available. As mentioned above, the lack of data on natural NNS would compromise the overall effect estimate of NNS intake during pregnancy on an offspring’s weight gain. Additionally, while all the included studies explored the artificial NNS, none specifically reported the ingredients of NNS. All the included studies were with a non-randomized design. Therefore, potential residual confounding and biases could not be fully controlled in observational studies, even though all the analyses were adjusted for known risk factors and potential confounders. The low quality of the body of evidence requires caution for data interpretation and more high-quality research for further explorations.

## 9. Conclusions

Maternal NNS intake during pregnancy was found to be associated with increased weight gain in offspring based on evidence from human studies. Further well-designed and adequately powered studies are needed to confirm this relationship.

## Figures and Tables

**Figure 1 nutrients-14-05098-f001:**
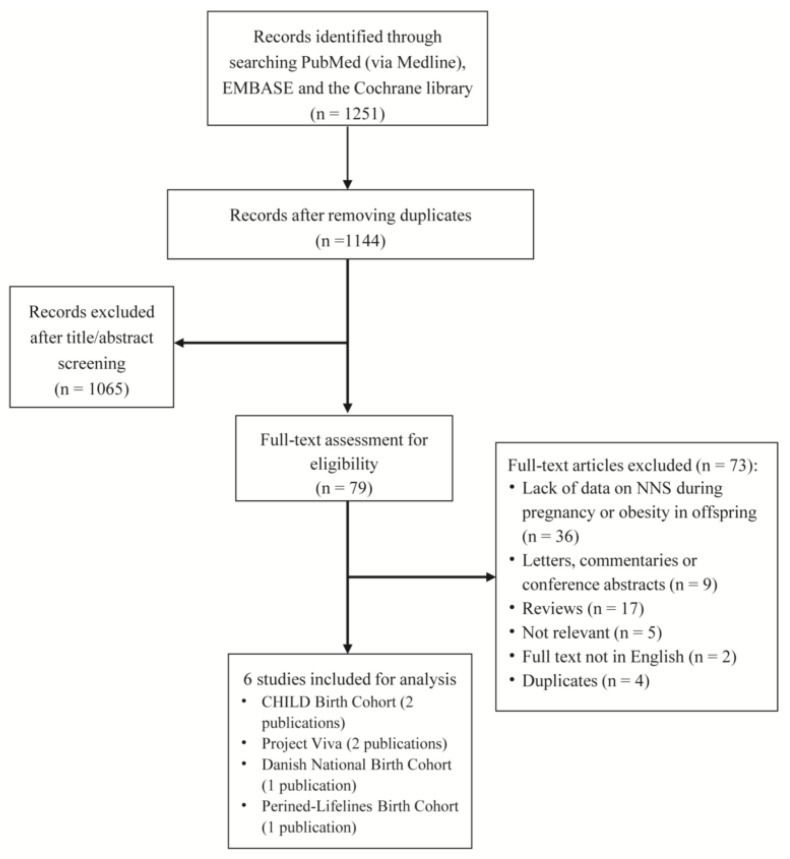
Flow diagram showing the study selection process.

**Figure 2 nutrients-14-05098-f002:**
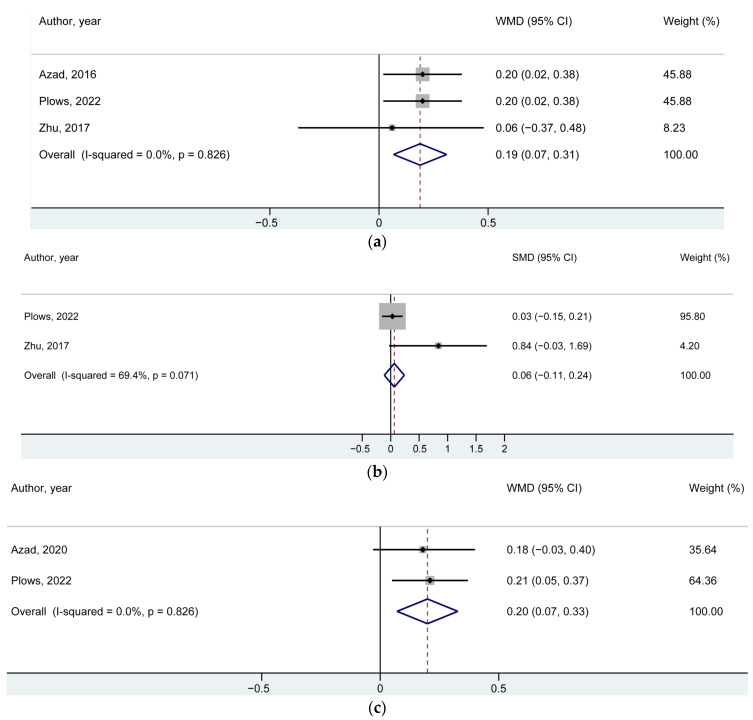
Forest plots for the relationship between NNS during pregnancy and BMI z-scores of offspring (**a**) for the outcome at 1 year of age; (**b**) for the outcome at birth; (**c**) for the outcome at early childhood; (**d**) for the outcome at mid-childhood; [6,7,15].

**Figure 3 nutrients-14-05098-f003:**
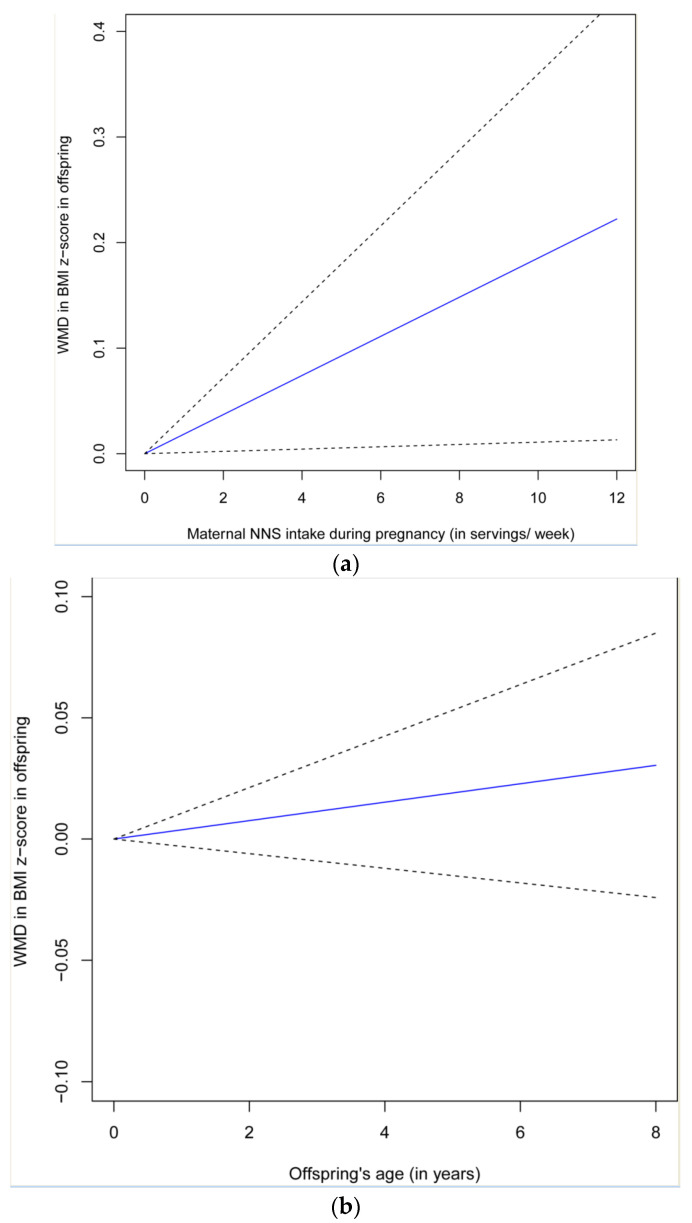
Dose-response relationship between NNS intake during pregnancy and WMD in BMI z-score ((**a**) for different NNS level in relation to WMD at 1 year of age; (**b**) for increased offspring’s ages in relation to WMD).

**Table 1 nutrients-14-05098-t001:** Characteristic descriptions of the included studies.

First Author, Publication Year	Country/Study Name	No. of Mothers/Offspring	Age of Mothers (in Years)	BMI of Mothers (in kg/m^2^)	Energy Intake of Mothers	Percentage of Mothers Smoking	Percentage of Gestational Diabetes	Girl Percentage of Offspring	Breastfeeding Duration for Offspring	Type of NNS	Measurement of NNS Intake	Outcomes Reported
Azad, 2016	Canada/CHILD Birth Cohort	2413/2413	32.5 (SD: 4.6)	24.8 (SD: 5.4)	2007 kcal/day	7.9%	4.4%	NA	8.8 months	Artificial	FFQ	Age- and sex-specific BMI-for-age z scores; offspring being overweight
Zhu, 2017	Denmark/Danish National Birth Cohort	918/918	31.3	27.6	2391 kcal/day	16.1%	100%	44%	NA	Artificial	FFQ	Age- and sex-specific BMI z-scores; offspring being overweight or obese; large-for-gestational age (LGA)
Salavati, 2020	The Netherlands/Perined-Lifelines Birth Cohort	1698/1698	29 (27–32)	23.8 (21.7–26.6)	1813 kcal/day	NA	NA	NA	NA	Artificial	FFQ	Birth weight
Plows, 2022	USA/Project Viva	1683/1683	32.2 (SD: 5.0)	24.6 (SD: 5.2)	NA	11%	5%	49%	NA	Artificial	FFQ	Age- and sex-specific BMI z-scores; sum of skinfolds; fat mass index

**Table 2 nutrients-14-05098-t002:** Relation between NNS and BMI z-score in offspring.

BMI z-Score	Number of Studies/Offspring	Reference Number for the Included Studies	WMD (95% CI)	*p*-Value
At 1 year of age	3/5014	[6,7,14]	0.19 (0.07, 0.31)	0.002
At birth	2/2601	[7,14]	0.06 (−0.11, 0.24) *	0.48
At early childhood	2/3981	[7,13]	0.20 (0.07, 0.33)	0.002
At mid-childhood	2/2601	[7,14]	0.29 (0.12, 0.46)	0.001

BMI = body mass index; WMD = weighted mean difference; CI = confidence interval. * Result shown as standardized mean difference (rather than WMD) because one of the included studies by Zhu et al., reported Ponderal Index for pooled analysis.

**Table 3 nutrients-14-05098-t003:** Some methodological issues identified from the included human studies.

Study Design Factor	Issue Descriptions
Population	· Lack of considerations for predefined subgroup explorations
Exposure	· No evidence for the natural NNS intake available · Insufficient data on quantitative NNS intake based on FFQ measurement · Failure to capture all beverages or foods with NNS for exposure assessment · Heterogeneity of the duration or time span for NNS intake during pregnancy
Control	· Heterogeneity of the definition for control group or the use of data-driven technique in defining control group
Outcome	· Heterogeneity of outcome definition and measurement · Insufficient information on accuracy of outcome assessment
Time	· Different time points of offspring in reporting relationship between NNS intake and offspring obesity

## Data Availability

Data presented in this manuscript are publicly accessible from the literature.

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
