# Peer review of "Consumption of Non-Nutritive Sweetener during Pregnancy and Weight Gain in Offspring: Evidence from Human Studies"

_nutrients, 2022, doi:10.3390/nu14235098_

Round 1

Reviewer 1 Report

The comparisonscriteria are not suitable

Author Response

Comment: The comparisons criteria are not suitable

Response: We thank the reviewer for his/her comment.

In this study, for the offspring’s outcomes, we compared their mothers between those consuming NNS versus those without NNS intake (i.e., ‘yes’ versus ‘no’) if the included studies reported results of this single pair-wise analysis, or between those consuming the highest level of NNS versus those with the lowest NNS intake level (i.e., ‘highest’ versus ‘lowest’) if the included studies reported data on multiple NNS intake groups. The latter comparison criteria were based on the Cochrane handbook recommendation (https://training.cochrane.org/handbook/current/chapter-23#section-23-3-4), aiming to avoid double-counting the participants in the shared control group and the unit-of-analysis errors (i.e., failing to account for the correlations between the multiple comparisons when comparing different higher NNS intake groups with the control group).

We have modified the sentence to enhance clarity (Lines 110-115 on Page 3). Thanks.

Reviewer 2 Report

Line 61-62, The electronic databases including PubMed (via Medline), EMBASE and the Cochrane Library were systematically searched until Sep 20th, 2022. There is no illustration for how large sample is?

Line 70-71, we chose the study with the largest sample size for inclusion. Line 139, We used data from the studies with the larger sample sizes for main analyses There is no illustration for how large sample is?

Line 114, p-value < 0.1 or the I2 > 50% indicated significant heterogeneity. How to explain that figure 2a and 2c I square 0.0%?

Table 1, Type of NNS, artificial, there is no further information of ingredients.

Line 145, As authors mention that “this study was the first to systematically explore the maternal NNS consumption during pregnancy in relation to offspring’s weight gain.” The manuscript’s title is interesting and not very new, but the authors only extract 6 articles, it seems too weak to illustration and not give us quate information.

Author Response

Comment 1: Line 61-62, The electronic databases including PubMed (via Medline), EMBASE and the Cochrane Library were systematically searched until Sep 20th, 2022. There is no illustration for how large sample is?

Response: We appreciate the reviewer’s comment. Revision has been made accordingly in the text (Lines 138-139 on Page 3).

Comment 2: Line 70-71, we chose the study with the largest sample size for inclusion. Line 139, We used data from the studies with the larger sample sizes for main analyses There is no illustration for how large sample is?

Response: Changes have been made based on the comment (Lines 140-144 on Page 3). Thanks.

Comment 3: Line 114, p-value < 0.1 or the I2 > 50% indicated significant heterogeneity. How to explain that figure 2a and 2c I square 0.0%?

Response: We have added details on the results of exploring heterogeneity in the text (Lines 4 and 8-9 on Page 6).

Comment 4: Table 1, Type of NNS, artificial, there is no further information of ingredients.

Response: We thank the reviewer for raising this comment. Unfortunately while all the included studies explored the artificial NNS, none specifically reported the ingredients of NNS. We emphasized this as a limitation in Discussion (Lines 164-165 on Page 10).

Comment 5: Line 145, As authors mention that “this study was the first to systematically explore the maternal NNS consumption during pregnancy in relation to offspring’s weight gain.” The manuscript’s title is interesting and not very new, but the authors only extract 6 articles, it seems too weak to illustration and not give us quate information

Response: We appreciate the reviewer’s suggestive comment.

In this study, we aimed to explore the maternal NNS consumption during pregnancy in relation to offspring’s weight gain, based on evidence from human studies. We found that the maternal NNS intake during pregnancy was significantly associated with increased weight gain in offspring. The study processes conducted in duplicate, in combination with the rigorous methodology, would strengthen the validity of study findings. However, we agreed with the reviewer about the lack of sufficient and solid evidence regarding the relationship between NNS intake and offspring’s weight gain. Our study also highlighted the need for further well-designed and adequately-powered research to clarify and validate the association, and to help translate the future synthesized evidence into public health policy-making and recommendation. We emphasized these points in the Discussion (Lines 149-154 on Page 10, Lines 172-173 on Page 11).

Reviewer 3 Report

The authors present a well-designed and complete systematic review of the risk of NNS during pregnancy, an important and interesting topic.  The study is particularly good in that 3 databases were searched.  The methods and statistics are thorough. 

My only concern is the authors do not make it clear whether there was any analysis possible between the aggregate NNS and control group for known/possible maternal risk factors such as socio-economic status, breastfeeding, gestational diabetes, mode of delivery, and maternal BMI.   Please address.  

Author Response

Comment 1: The authors present a well-designed and complete systematic review of the risk of NNS during pregnancy, an important and interesting topic. The study is particularly good in that 3 databases were searched. The methods and statistics are thorough. 

Response: We appreciate the reviewer’s positive comments on our manuscript.

Comment 2: My only concern is the authors do not make it clear whether there was any analysis possible between the aggregate NNS and control group for known/possible maternal risk factors such as socio-economic status, breastfeeding, gestational diabetes, mode of delivery, and maternal BMI. Please address.

Response: We would like to thank the reviewer for his/her helpful comment.

We tried to collect the adjusted relationship between NNS and offspring’s weight gain from the published human studies. However, the covariate adjustment in the models may not be the same as from all the included studies, whilst some studies may fail to account for the important maternal risk factors including socio-economic status, breastfeeding, gestational diabetes, mode of delivery, and maternal BMI. Unfortunately, we could not perform these subgroup analyses stratified by the maternal risk factors due to the insufficient included studies and limited data available.

We have emphasized this in the Discussion (Lines 156-162 on Page 10). Thanks.

Round 2

Reviewer 1 Report

WELL WRITTEN